# Fluorescence Spectroscopy of Low-Level Endogenous β-Adrenergic Receptor Expression at the Plasma Membrane of Differentiating Human iPSC-Derived Cardiomyocytes

**DOI:** 10.3390/ijms231810405

**Published:** 2022-09-08

**Authors:** Philipp Gmach, Marc Bathe-Peters, Narasimha Telugu, Duncan C. Miller, Paolo Annibale

**Affiliations:** 1Max Delbrück Center for Molecular Medicine in the Helmholtz Association, 13125 Berlin, Germany; 2Institute of Pharmacology and Toxicology, University of Würzburg, 97070 Würzburg, Germany; 3German Centre for Cardiovascular Research, Partner Site Berlin, 10785 Berlin, Germany; 4School of Physics and Astronomy, University of St Andrews, St Andrews KY16 9AJ, UK

**Keywords:** GPCR, β-adrenergic receptors, hiPSC-CM, cardiomyocyte, fluorescence correlation spectroscopy, FCS, fluorescence, CRISPR/Cas9, differentiation

## Abstract

The potential of human-induced pluripotent stem cells (hiPSCs) to be differentiated into cardiomyocytes (CMs) mimicking adult CMs functional morphology, marker genes and signaling characteristics has been investigated since over a decade. The evolution of the membrane localization of CM-specific G protein-coupled receptors throughout differentiation has received, however, only limited attention to date. We employ here advanced fluorescent spectroscopy, namely linescan Fluorescence Correlation Spectroscopy (FCS), to observe how the plasma membrane abundance of the β_1_- and β_2_-adrenergic receptors (β_1/2_-ARs), labelled using a bright and photostable fluorescent antagonist, evolves during the long-term monolayer culture of hiPSC-derived CMs. We compare it to the kinetics of observed mRNA levels in wildtype (WT) hiPSCs and in two CRISPR/Cas9 knock-in clones. We conduct these observations against the backdrop of our recent report of cell-to-cell expression variability, as well as of the subcellular localization heterogeneity of β-ARs in adult CMs.

## 1. Introduction

Human-induced pluripotent stem cells (hiPSCs) can be differentiated into cardiomyocytes (CMs) that are functionally comparable to embryonic stem (ES) cell-derived CMs and/or mimicking the functional morphology of adult cardiomyocytes. Although fully differentiated hiPSCs do not yet achieve a phenotype entirely reminiscent of that of adult CMs—an area of active and intense research [1]—the possibility of driving cells in a controlled way these from a pluripotent towards a more mature phenotype offers several important unique opportunities. First and foremost, undifferentiated hiPSCs are amenable to genetic editing, which allows them to incorporate the knockouts of selected genes or addition of functional tags, such as fluorophores, that will then be carried forward to the CM phenotype.

Secondly, the expression of key genes changes over time and this can be followed in detail, providing an insight into how key structures of the CM phenotype originate, and how specific signaling pathways and their associated molecular mechanisms develop. In this respect, several works have explored since about 10 years the evolution of the transcriptome of hiPSCs as they directionally differentiate towards CMs. The majority of CM-specific genes seem to be upregulated after day 14 [2,3]. More recently, single cell sequencing has allowed light to be shed also on cell-to-cell expression variability for selected genes [4,5].

An important feature of adult CMs is the ability to increase or reduce both the strength and frequency of their contractions in response to external stimuli, two processes that go under the name of inotropic and chronotropic responses, respectively. The β-adrenergic system, embodied by the β_1_- and β_2_-adrenergic receptors (β_1/2_-ARs, expressed from the *ADRB1* and *ADRB2* genes, respectively), represents the typical pathway mediating catecholamine-driven inotropic and chronotropic responses, and is a crucial therapeutic target in the context of heart failure, mostly thanks to the discovery of the beneficial action of negative inotropes/chronotropes, such as β-blockers. In the adult heart, β_1_ and β_2_ are both expressed with a ratio of β_1_/β_2_-ARs of around 70–80%/30–20% in human ventricles [6,7].

The evolution of endogenous β-AR expression over maturation time in hiPSCs directionally differentiated towards CMs has thus been explored in several recent works. Wu et al. observed mRNA levels and effects on the contractility of β-AR expression in differentiating hiPSC, in a model of dilated cardiomyopathy [8]. Jung et al. studied adrenoreceptor mRNA and total protein levels within the context of functional remodeling [9], while Hasan et al. examined how receptor-specific cAMP signaling evolves throughout hiPSC differentiation towards CMs [10]. Kondrashov et al. studied the temporal dynamics of β_2_-AR expression in CRISPRed hiPSC [11]. An earlier work by Yan et al. studied the effect of adrenergic signaling on regulating the cardiac differentiation of mouse embryonic stem cells and monitored *ADRB1* and *ADRB2* expression as a function of time [12].

These works interestingly report a predominant *ADRB2* expression with respect to *ADRB1*, resulting in a higher second messenger production upon selective β_2_-AR stimulation, that then decays during differentiation. This results in a β_2_-AR-dominated functional response (contractility), which is then caught-up by the β_1_-AR at day 60. These results point to the interesting and persistent gene expression dynamics affecting the relative weight of these two key adrenoreceptors. However, transcriptomic analyses and functional downstream readouts, even if conducted at the single cell level, fail to provide information about spatially resolved receptor expression. We believe this is an important aspect to consider in the light of our recent observation of the peculiar differential localization of the β_1_- and β_2_-ARs on the membranes of adult cardiomyocytes [13]. We were able to show that the β_2_-AR partitions to the T-tubular network of the cardiomyocyte and does not appear to diffuse upon the basolateral (‘crest’) surface plasma membrane of the cell, thus explaining the compartmentalized cAMP signal observed earlier in this cell-type [14].

Key to our observation was the use of a high-sensitivity fluorescence spectroscopy method, namely linescan-FCS [15], that allowed us to appreciate the minute amounts of endogenous receptors diffusing on the membranes using a fluorescent antagonist (JE1319). In that context we were also able to observe not only the significant CM to CM variability of receptor expression (i.e., β_1_-AR is visible only in 60% of the cells considered), but also the fact that, in CM-like cells such as H9c2 and hiPSC-CMs, β_2_-ARs were clearly seen diffusing on the basolateral membrane.

We thus reasoned that it is now timely to investigate how the plasma membrane concentration of the β_1_- and β_2_-adrenergic receptors evolves during long-term monolayer culture of hiPSC-CM, whilst comparing it to the measured mRNA levels. Moreover, considering that one of the key advantages of hiPSCs is their ability to be gene-edited in their pluripotent, non-differentiated state, we also extended our analysis to two hiPSC clones carrying a heterozygous and a homozygous CRISPR/Cas9-mediated knock-in of a fluorescent tag at the N-terminus of the *ADRB2*. This should further allow the corroboration of the missing link between protein localization and mRNA levels.

We could observe that in wildtype (WT) hiPSCs, there is a discrepancy between β_2_-AR appearance at the surface plasma membrane during early maturation stages (day 25) with respect to the measured mRNA levels, and a significant cell-to-cell heterogeneity in receptor abundance at the plasma membrane. At later maturation stages (day 60) mRNA levels correlated with the membrane appearance of β_2_-AR in WT hiPSC-CM as well as in CRISPRed hiPSC-CM clones, indicating that the addition of a tag at the N-terminus of *ADRB2*/β_2_-AR still allowed for the correct receptor transcription, expression and localization at and after day 60.

These observations open fascinating questions on the mechanisms that control and time receptors appearance and distribution at the plasma membrane, with the potential implication towards a better understanding of a signaling cascade at the core of heart pathophysiology and failure.

## 2. Results

### 2.1. Generation and Characterization of WT and CRISPRed hiPSC-CM

We first set to establish a CM-differentiation assay using our hiPSC lines (See Materials and Methods Section and Appendix A). Preliminary experiments in the parental WT (BIHi005-A) line indicated that our protocol efficiently specified hiPSC-CMs expressing cardiac Troponin T, which increased in ventricular-specific MLC2v expression following dissociation and replating between weeks two and three (Appendix A). Through this we identified timepoints (0, 25, 60 and 100 days) after the onset of differentiation for our gene expression and spectroscopic investigation (Appendix A). The hiPSCs morphology was investigated by transmitted light microscopy, as reported in Figure 1A and Appendix A.

We worked with three hiPSC lines—the parental WT hiPSCs, two clonal lines arising respectively from a heterozygous (66–35) and the homozygous knock-in (16–31) of the monomeric Enhanced Green Fluorescent Protein (mEGFP) at the N-terminus of the endogenous *ADRB2* gene locus, preceded by the membrane targeting signal sequence previously reported by Guan et al. [16,17]. Clones 16–31 and 66–35, created using CRISPR/Cas9 editing in combination with a novel partially single-stranded, DNA-based hybrid HDR donor cocktail, and details of the knock-in procedure as well as the clones’ identification are described in Appendix A and the Methods. The clones retained their pluripotency after gene editing as revealed by their morphology as well as by flow cytometry (Appendix A).

Figure 1A further displays the fluorescence excitation of the cells at 488 nm, indicating a substantial amount of autofluorescence [18,19], which increases as cells differentiate towards a mature CM phenotype. The primary source of autofluorescence appears to be the abundant mitochondrial networks present in the differentiated hiPSC-CMs (Appendix A). Interestingly, the level of green signal is comparable between the WT cells and the two mEGFP knock-in clones, suggesting that the very low-level expression of the tagged β_2_-AR has to compete with autofluorescence and that alternative tagging strategies shall be considered [20,21]. In the following, we therefore employed the fluorescent antagonist, which we successfully used in our recent spectroscopic imaging of adrenergic receptors in adult murine cardiomyocytes (JE1319 [13]), while still exploiting the information arising from the two knock-in clones.

Figure 1B displays the fully differentiated hiPSC-CM phenotype at day ~100, employing α-actinin and cardiac troponin T immunofluorescence to highlight the presence of sarcomeric structures (sarcomere length of ~2 µm) and thus the success of differentiation. This shall be compared to the lower immunofluorescence signal observed in the few non-CM cells present in the culture (dashed lines in the DIC and Merge overlay of Figure 1B). Morphologically, there are no detectable differences between the WT cells and clones 16–31 and 66–35.

When subjected to selective adrenergic stimulation (for either β_1_-AR using ICI-118551 or β_2_-AR using CGP-20712A) (Figure 1C), the day ~100 hiPSC-CM displayed a chronotropic response, as measured by their beating rate, comparable between the WT cells and the homozygous clone 16–31. In particular, the selective β_2_-AR stimulation displayed an increase in beating rate almost identical between the WT and the knock-in, suggesting that the N-terminal tagging of the β_2_-AR neither affects its proper expression nor its function. This is also displayed in Videos S1 and S2, which show the spontaneously contracting monolayers of the day 100 clone 16–31 before and after β_2_-AR stimulation, respectively. These results also confirm that the almost undetectable levels of expressed receptor can contribute to the normal response of the cell to external stimuli [13].

### 2.2. Evolution of ADRB2 Transcript Levels during Differentiation and Maturation

Having established and characterized our three hiPSC-CM lines, we now set out to explore in detail the kinetic of β_2_-ARs expression as a function of the maturation time. In order to do so, we first performed quantitative mRNA measurements using qRT-PCR (see Materials and Methods Section). Here, the presence of a tag in the two knock-in clones further allows to measure the expression levels of individual alleles, as well as to compare transcript levels from a tagged variant of the β_2_-AR to its WT expression. The results of this experiment are displayed in Figure 2: here, we compare absolute mRNA levels (Figure 2A) extracted from cultures of WT hiPSC and our two clones. Two sets of qPCR primers, as detailed in the methods, one complementary to the mEGFP sequence (green bars) and the other complementary to the *ADRB2* nucleotide sequence (black bars) were used, thus allowing the comparison of the expression level of the endogenous background with respect to the tagged allele(s) (see Materials and Methods Section).

Upon normalization to *GAPDH* and the day 0 expression levels of the homozygous clone 16–31, Figure 2B allows us to fully appreciate the temporal evolution of *ADRB2* transcripts amount along the differentiation pathway. In accordance with what was observed in previous reports [8], *ADRB2* mRNA transcripts appear to be substantially upregulated already after day 25 from the onset of differentiation, with modest increases along the further timepoints. As expected, clone 16–31 displays comparable expression levels of mRNA containing the *mEGFP* sequence and of mRNA containing the *ADRB2* coding sequence, indicative that both alleles carry the knock-in. Instead, clone 66–35 displays roughly half *mEGFP* mRNA than *ADRB2* mRNA, indicative for both a monoallelic knock-in as well as of a roughly equal level of expression of both alleles throughout the differentiation. In the light of these observations, we here ascribe the lower overall *ADRB2* mRNA levels observed in both clone 16–31 and clone 66–35 to clonal variability.

### 2.3. β-Adrenergic Receptor Expression at the Plasma Membrane of Differentiating hiPSC-CM

The next question is now if and how mRNA expression levels of the *ADRB2* over time are mirrored by the β_2_-AR expression levels at the plasma membrane. In order to measure the latter, we resort to the approach we already successfully used to study β-AR expression in adult CMs, based on linescan-FCS to monitor the minute amounts of diffusing receptors tagged with our fluorescently labeled antagonist. Briefly, the excitation volume of the confocal microscope is scanned rapidly (<1 ms) along a line, positioned along the basolateral membrane of the cell. Fluctuations due to the diffusion of the tagged receptors are extracted by a mathematical analysis known as autocorrelation in order to yield a curve termed autocorrelation function (ACF) that contains information both on the diffusivity and concentration of the tagged receptors. This function can be then displayed in a color code and an average ACF from several cells pooled in a heatmap image that allows direct comparison between and across clones and timepoints. In general, the qualitative observation of an ACF, with decaying amplitude (from red—high correlation—to blue—no correlation) indicates the presence and diffusion of labeled receptors. For this reason, the method can be used in a powerful way to estimate the presence of the β_1_-AR and β_2_-AR at the basolateral membrane in our hiPSC-CMs and measure their evolution over time. The result of this analysis is displayed for the WT hiPSC-CM in Figure 3, comparing β_1_-AR and β_2_-AR expression over time. Most of the undifferentiated (day 0) cells display low to negligible level of β_1_-AR and β_2_-AR expression, with the exception of one cell where β_2_-AR was selectively labeled. Along these lines the day 25 cells investigated still displayed a substantial cell-to-cell heterogeneity in plasma membrane expression, with a significant amount of cells displaying no trace of either β_1_-AR (~90%) or β_2_-AR (~75%) at the cell membrane. This is a striking departure from the mRNA data reported in Figure 2, as the single cell plasma membrane receptor abundance data paint a significantly more complex and heterogeneous picture than the bulk mRNA levels. Both receptors become visible at the plasma membrane in most cells at 60 days from the onset of differentiation, and no qualitative difference is observed between day 60 and day 100. These results open an interesting paradox, since this would imply that at 25 days post differentiation the cell has substantial amounts of *ADRB2* mRNA but still no comparable levels of β_2_-AR expressed at the plasma membrane, pointing to either a limitation on translation of the receptor mRNA or of its correct targeting, as cells progress at different paces through the differentiation pathway.

We further extracted the diffusion constants (found in Appendix A) and mean values range from 0.25–0.45 µm^2^/s. Appendix A gives an indication of the observed receptor concentrations, with mean values ranging from 3–12 particles in the effective detection volume of the confocal microscope. In this case, β_2_-AR expression appears higher than β_1_-AR both at day 60 and day 100.

Finally, we asked whether the knock-in of an N-terminal tag, which did not seem to affect mRNA levels, ended up influencing proper receptor localization at the plasma membrane. The construct corresponding to our knock-in modification of the *ADRB2* appears to be well expressed and clearly localized to the plasma membrane when transfected as a plasmid into H9c2 cells (Appendix A). Figure 4 displays, for day 60 (when the full expression of the receptor on the plasma membrane is observed) that membrane expression levels are comparable between WT hiPSC-CM and the two knock-in clones, with the possible exception of β_2_-AR membrane expression on the homozygous clone (16–31). Reassuringly, no membrane signal compatible with receptor diffusion was observed in the control, meaning that cells treated with the inverse agonist propranolol before being exposed to the fluorescent ligand did not display any ACF (Figure 4C).

The data from β_2_-AR on day 60 of clone 16–31, display visible receptor presence at the membrane and dynamics, at least for three of the cells, whereas the other four cells do not yield a signal compatible with receptors diffusing at the membrane. Taken together with the observation of the sizable β_2_-AR-mediated increase in contractility (Figure 1C), we believe that clone 16–31 is properly expressing the tagged β_2_-ARs to the plasma membrane, albeit displaying higher cell-to-cell variability.

## 3. Discussion

Our work has quantitatively explored the membrane expression level of the β_1_-AR and β_2_-AR in differentiating hiPSCs at four time points, i.e., in the undifferentiated pluripotent state, and then 25, 60 and 100 days after directed differentiation towards a CM state. Our work was conducted in parallel on WT hiPSCs, together with two CRISPR/Cas9 knock-in clones, incorporating, respectively, a homozygous and heterozygous N-terminal tag (mEGFP) at the β_2_-AR, allowing the monitoring of the effect of a tag on the transcription of the *ADRB2* gene, as well as on the expression and localization of the β_2_-AR.

Membrane protein abundance, measured by linescan-FCS, a sensitive fluorescence spectroscopy approach, was then compared to mRNA levels obtained from qPCR. It shall be noted that antibodies raised against GPCRs have well-documented shortcomings, and even when using anti-EGFP antibodies against the N-terminal tag in clones 16–31 and 66–35, we failed to detect receptor expression in Western blots, reaffirming the necessity of a high-sensitivity method, such as our linescan-FCS technique.

The approach we implemented allowed us to observe a significant discrepancy between the mRNA levels of the β_2_-AR expression at the plasma membrane 25 days after differentiation, which raises interesting questions concerning translational regulation mechanisms for *ADRB1/2* in differentiating hiPSC-CMs. Our work fills a gap in the existing literature, which either addressed only mRNA levels, or functional readouts after day 30 of differentiation in hiPSCs. So far, only two works, one on mouse ES cells and one on hiPSC-CMs, measured protein expression levels as a function of differentiation, further extracting total cell levels and not being able to resolve membrane-specific expressions [9,12]. The relationship between mRNA levels and protein levels in mammalian cells is complex, and although a general correlation is expected, also making transcriptional control the dominant mechanism regulating protein abundance, there are several exceptions to this rule, as reviewed by Liu et al. [22]. In differentiating cells, the situation is even more complex, and translational control has been indicated to play a major role, especially in the early stages of differentiation [23]. We shall note here that our method allows only the monitoring of the receptor expression levels at the basolateral membrane, since we use a ligand that is non permeable to the plasma membrane [13], and we thus cannot rule out that the β_2_-AR is still translated but does not localize to the plasma membrane. It could be that accessory proteins responsible for its trafficking or posttranslational modifications are not yet present in the differentiating cell at day 0 and day 25 [24]. It shall be further noted that the fully differentiated hiPSC-CMs, even at day ~100, do not display a complete T-Tubular network, leaving open the question if the β_2_-AR, that in this work we observed on the basolateral membrane, would eventually segregate in these compartments [13]. Our observation of higher relative β_2_-AR expression levels in hiPSC-CMs appears to be in line with what was previously reported in at least three works [8,9,10]. However, we shall mention here that in recent measurements in hiPSC-derived engineered heart tissue, higher levels of β_1_-AR were observed throughout (private communication and [25]).

We shall also note here that our choice of mEGFP as an N-terminal tag for the live cell imaging of the β_2_-AR, while well-established in overexpression systems, faced significant challenges due to the high autofluorescence background of differentiated hiPSC-CMs, as displayed in Figure 1A. Nevertheless, the use of the tag was instrumental in confirming that the effect of the addition of a ~700 bp sequence to the *ADRB2* gene, resulting in a ~25 kDa tag on the final protein, did not perturb its transcription, its translation nor its final plasma membrane localization.

Overall, we can show here that linescan-FCS can be successfully used to monitor the endogenous low level membrane receptor expression of GPCRs in differentiating hiPSC-CM lines, as recently demonstrated for single point FCS by Goulding et al. in HEK293 cells [20]. This substantially increases the possibilities of current methods available for the assessment of gene expression throughout hiPSC differentiation: not only we add to quantitative mRNA measurements also the dimension of actual protein expression levels, but furthermore we show how to measure it in a spatially resolved way within a single cell.

## 4. Materials and Methods

### 4.1. Oligonucleotides

A list of Oligonucleotides used in this work is displayed in Table 1.

### 4.2. Differentiation and Culture of hiPSC-CM

In this work we used a well-established hiPSC line (BIHi005-A) of the stem cell core facility of the Max Delbrück Center for Molecular Medicine, which was obtained by reprogramming of human dermal fibroblasts using Sendai viral vectors, as previously reported (refer to European Human Pluripotent Stem Cell Registry (hPSCreg) database for details https://hpscreg.eu/cell-line/BIHi005-A (accessed on 24 August 2022)). Pluripotent cultures were grown at 37 °C with 5% CO_2_ and 5% O_2_, whereas differentiated cultures were maintained at 5% CO_2_ and atmospheric (21%) O_2_. The monolayers of hiPSCs were differentiated into hiPSC-CMs by modulating Wnt signaling according to a small molecule-based cardiac differentiation strategy [26,27], followed by metabolic lactate selection, in order to enrich for cardiac myocytes [28], as also previously described by us [13,29]. In short, hiPSC were cultured in essential 8 basal medium (E8 medium) (Gibco, Waltham, MA, USA) on Geltrex (Gibco)-coated plates 4 days prior to cardiac induction. When they reached 80–90% confluency, mesodermal differentiation was induced on day 0 for 24 h by changing the medium to cardiac priming medium (RPMI 1640 (Gibco), 1× B27 supplement without insulin (Gibco) and 6 µM CHIR99021 (Sellek Chem, Houston, TX, USA)). On day 1, basal differentiation medium (cardiac priming medium without CHIR99021) was provided on top of the old cardiac priming medium. Two days later cardiogenesis was induced by changing the medium to basal differentiation medium supplemented with 5 µM of the Wnt inhibitor IWR-1-endo (Sellek Chem). On day 5, basal differentiation medium was added on top of the old medium to let cells grow for another 2 days. Cells were cultured in maintenance medium (RPMI 1640, 1× B27 with insulin (Gibco)) starting at day 7. The selection of and enrichment for cardiomyocytes was initiated on day 9 by culturing cells for 4 days in RPMI 1640 medium without glucose (Gibco), supplemented with 213 µg/mL L-ascorbic acid 2-phosphate (Sigma-Aldrich, Saint Louis, MO, USA) and 500 µg/mL Human Recombinant Albumin (Sigma-Aldrich) (for further details we refer the reader to CDM3 medium described in [26]) and 5 mM sodium DL-lactate (Sigma-Aldrich). From here, the beating hiPSC-CMs were cultured in maintenance medium, which was exchanged every 3–5 days. This protocol efficiently generates cardiac Troponin T-positive hiPSC-CM populations in two weeks after the onset of differentiation (>95%, as in Appendix A). Furthermore, during week 2 and 3, the hiPSC-CM population increasingly becomes positive for the ventricular marker MLC2v. We observed a batch to batch variability of cells expressing MLC2v at day 20 ranging from 18% to 48%.

### 4.3. Culture of H9c2 Cells

H9c2 (ATCC; CRL-1446) cells were maintained in Dulbecco’s modified Eagle’s medium (PAN-Biotech, Aidenbach, Germany) supplemented with 10% (*vol*/*vol*) fetal bovine serum (FBS) (Sigma-Aldrich, Saint Louis, MO, USA), 2 mM L-glutamine (PAN-Biotech, Aidenbach, Germany), penicillin (100 U/mL; Gibco) and streptomycin (100 μg/mL; Gibco) at 37 °C and 5% CO_2_. Cells were passaged and dissociated using 0.05%/(ethylenedinitrilo)tetraacetic acid (EDTA) 0.02% in phosphate buffered saline (PBS) (PAN-Biotech).

### 4.4. CRISPR Strategy

Efficient endogenous mEGFP-tagging at the N-terminus of *ADRB2* in hiPSC was achieved using CRISPR/Cas9 and partially single-stranded, dsDNA-based hybrid homology-directed repair (HDR) donors with symmetric homology arms as previously described [30]. First, a sequence-verified HDR donor plasmid was designed and synthesized (Genscript, Piscataway, NJ, USA), including the mEGFP coding sequence flanked by 230 bp homology arms (refer to “Section 4.1, Oligonucleotides”). Two different primer pairs were used to generate two PCR products from this donor plasmid encoding for mEGFP alone and for mEGFP plus 120 bp 5′- as well as 3′-homology arms, respectively (refer to “Oligonucleotides”). A subsequent melt and anneal reaction led to the generation of the partially single-stranded hybrid donor cocktail with HDR donors containing symmetric homology arms (Appendix A). This cocktail was provided to the cells together with a ribonucleoprotein complex consisting of crRNA, tracrRNA and Cas9 protein (IDT, refer to “Oligonucleotides” for crRNA sequence) via nucleofection using a 4D nucleofector (Lonza). Next, we generated single cell clones by automated hiPSC single cell seeding and clonal expansion using the iotaSciences IsoCell platform [31]. We screened for mEGFP knock-in at the *ADRB2* locus by PCR genotyping and selected one homozygous (16) and one heterozygous (66) clone for further experiments.

A second CRISPR round was performed on the two *mEGFP-ADRB2* CRISPRed clones, 16 and 66, in order to introduce the influenza hemagglutinin signal sequence (HA-SS) (MKTIIALSYIFCLVFA [16]) at the N-terminus of *mEGFP-ADRB2*. For this, a single-stranded oligodeoxynucleotide (ssODN) encoding for the HA-SS as well as 60bp homology arms was designed and synthesized (IDT) and provided to the cells, together with a novel crRNA that specifically targeted the *mEGFP-ADRB2* locus. Again, single cell clones were generated as described above and screened for homozygous (16–31) and heterozygous (66–35) hiPSC clones via restriction fragment length polymorphism (RFLP) based on the BbsI restriction enzyme cut site introduced by the HA-SS.

### 4.5. Flow Cytometry

HiPSCs and hiPSC-CM were dissociated as single cells using TrypLE™ Select Enzyme (1x) (Gibco) and TrypLE™ Select Enzyme (10x) (Gibco), respectively. For hiPSCs 2 × 10^5^ cells were fluorescently labelled for pluripotent stem cell surface markers using 1:20 anti-SSEA4-VioBlue (Miltenyi Biotec, Bergisch Gladbach, Germany) 130-098-366) and 1:600 anti-Tra1-60-Vio488 (Miltenyi Biotec, 130-106-872) antibodies. Then, cells were fixed and permeabilized using the FoxP3 Staining Buffer Set (Miltenyi Biotec, 130-093-142) and labelled for nuclear markers using 1:50 anti-Oct3/4-APC (Miltenyi Biotec, 130-123-318) and 1:100 anti-Nanog-PE (Cell Signaling, Danvers, MA, USA, 14955S). For the characterization of cardiac markers 2 × 10^5^, hiPSC-CMs were fixed and permeabilized similarly to the hiPSCs and fluorescently labelled using 1:50 anti-cardiac Troponin T-FITC (Miltenyi Biotec, 130-119-575) and 1:10 anti-MLC2v-APC (Miltenyi Biotec, 130-106-134) antibodies. Isotype stainings were performed as controls. Cells were measured for marker expression using a MACSQuant VYB (Miltenyi Biotec) flow cytometer and data analyzed using FlowJo software.

### 4.6. Contractility Assay

Cells were imaged in 6-well dishes in brightfield mode on a custom-built Zeiss setup. Time series videos of 3–5 min length (with 5 frames per second) were acquired before treatment, after antagonist incubation (100 nM ICI-118551, 300 nM CGP-20712A and 10 µM propranolol) and within 30 min after agonist treatment (1 µM isoproterenol). Contraction rates and other parameters were extracted from videos using the imageJ software “Myocyter v1.3” [32].

### 4.7. qRT-PCR

Cells were lysed and RNA extracted according to the RNeasy Mini kit (Qiagen, Hilden, Germany). The cDNA synthesis was performed using the SuperScript III First-Strand Synthesis SuperMix (Invitrogen, Waltham, MA, USA). Finally, qRT-PCR was performed using the QuantiTect SYBR Green PCR kit (Qiagen) on a QuantStudio 6 flex qPCR device (Thermo Fisher Scientific, Waltham, MA, USA). Primers for *GAPDH* (Geneglobe ID: QT00079247) and *ADRB2* (Geneglobe ID: QT00200011) were purchased from Qiagen and primers for *mEGFP-ADRB2* were self-made (refer to “Oligonucleotides”).

### 4.8. Cell Preparation for Confocal Imaging

Three days prior to imaging hiPSCs as well as hiPSC-CMs were detached as single cells and seeded in Geltrex-coated, eight-well, glass-bottom µ-slides (Ibidi, Gräfelfing, Germany). Until the day of experiment hiPSC-CMs were cultured in a maintenance medium with 3 µM CHIR99021 and 1x RevitaCell supplement (Life Technologies, Carlsbad, CA, USA). H9c2 cells were seeded in eight-well glass-bottom μ-slides (Ibidi) without coating and transfected 24 h later using Lipofectamine 2000 (Thermo Fisher Scientific) according to the manufacturer’s instructions. Two days post transfection H9c2 cells were used for imaging. All cells in this study (H9c2, hiPSC and hiPSC-CM) were cultured in the same imaging buffer (pH 7.4, 20 mM 4-(2-hydroxyethyl)-1-piperazineethanesulfonic acid [Hepes], 137 mM NaCl, 5 mM KCl, 1 mM MgCl_2_, 1 mM CaCl_2_ and 0.5% bovine serum albumin [BSA]) during microscopic analysis. Prior to imaging, hiPSC-CMs were preincubated for 40 min to 1 h with either 100 nM CGP-20712 (to image β_2_-AR) or 50 nM ICI-118,551 (to image β_1_-AR) diluted in the imaging buffer. Then, 50 nM JE1319 ligand was diluted in imaging buffer and directly added to the cells (with CGP-20712 or ICI-118,551 as described) for 40 min to 1 h. During the last 15 min of ligand incubation, 0.25× CellMask Green Plasma Membrane Stain (Invitrogen) was added to the hiPSC-CMs. After all incubations, hiPSC-CMs were washed three times using imaging buffer and were imaged in imaging buffer containing the respective β-AR antagonist as well as 50 μM para-aminoblebbistatin (Optopharma) to inhibit spontaneous contractions. Finally, cells were imaged on a Leica SP8 WLL confocal laser scanning microscope under physiological conditions (37 °C, 5% CO_2_, 85% humidity) using a sample incubator (Stage Top Chamber; OKOlab, Pozzuoli, Italy).

### 4.9. Immunofluorescence

At the day of fixation hiPSC-CM were washed with 250 µL phosphate buffered saline (PBS) (PAN-Biotech) per 8-well µ-slide (Ibidi). Cells were fixed using 4% paraformaldehyde for 30 min at room temperature (RT). Afterwards, cells were washed again with PBS and stored at 4 °C. Cells were blocked using NDS buffer (10% NDS, 0.2% BSA, 0.3% TritonX100) for 1 h at RT and directly labelled overnight at 4 °C with primary antibodies (1:400 mouse mAB for sarcomeric alpha-actinin (CloneEA-53, Sigma-Aldrich, A7811) and 1:200 rabbit pAB for cardiac Troponin T (Abcam, ab45932)). After washing cells once with PBS, secondary antibodies were applied for 2 h in the dark (1:250 anti-rabbit IgG coupled to AlexaFluor-555 (Invitrogen, A31572) and 1:250 anti-mouse IgG coupled to AlexaFluor-488 (Invitrogen, A21202)). Two drops of NucBlue™ Fixed Cell ReadyProbes™ (Invitrogen, R37606) were added to the secondary Ab solution to stain for nuclei. Cells were washed twice with PBS and then imaged on a Leica SP8 WLL.

### 4.10. Acquisition of Linescans

In order to extract the behavior of the diffusing species, linescan-FCS (Fluorescence Correlation Spectroscopy) was performed, in which a confocal beam is repeatedly scanned at high speed over the same portion of the sample. The resultant recorded kymographs or linescans are further analyzed by a custom-implemented code in MATLAB to extract information about the diffusion data of receptors from autocorrelation functions (as previously described in [15]).

In short, linescans were acquired on a commercial confocal laser scanning microscope, Leica SP8 WLL, with a white light laser (WLL). Via the HC PLAP CS2 40x 1.3 NA oil immersion objective (Leica) a pixel size of 50 nm was used to acquire about 6 × 10^5^ lines of 256 pixels each at a speed of 1800 Hz. In order to stabilize the focal position, the IR laser-based autofocus of the microscope (Leica, Adaptive Focus Control) was enabled, while the ligand JE1319 (Alexafluor 647-based) was excited at a wavelength of 633 nm with 5% or 50% laser power, upon which calibration corresponded to total power outputs in the µW-range. Hybrid detectors (HyD) in photon counting mode were applied to detect emissions in the range of 650–751 nm. The beam waist was extracted by the observation of the profiles of fluorescent microspheres (Tetraspeck, Thermo Fisher Scientific), as in [15], resulting in a lateral waist of ω_0_^(633 nm)^= 0.33 µm.

### 4.11. Analysis of Confocal Linescans

Confocal linescans were analyzed as previously described [13,15]. Briefly, fluorescence correlation spectroscopy (FCS) extracts information on equilibrium processes in the sample by comparing the statistical fluctuations of fluorescent particles in an effective detection volume. Linescan-FCS greatly increases the statistical accuracy of these measurements due to the repeated scanning of one line. The recorded time-trace of fluorescent intensity (I) fluctuations in the effective detection volume allows for the calculation of an autocorrelation G(τ) reflecting the timescale of these fluctuations.
(1)$$G\left(\tau \right)=\frac{I\left(t\right)\cdot I\left(t+\tau \right)}{I{\left(t\right)}^{2}}$$

The pointed brackets represent the average over all time values t. In order to extract diffusion parameters, such as the diffusion time τD or the diffusion constant D (which are connected via τ_D_ = $\frac{\omega_{0}^{2}}{4D}$), the autocorrelation function G(τ) is fit to the two-dimensional model of the autocorrelation function.
(2)$$G_{2D}\left(\tau \right)=\frac{1}{N}\frac{1}{1+\frac{4D\tau }{\omega_{0}^{2}}}+{\;G}_{\infty }$$
where τ represents the time lag, N the average number of particles and $G_{\infty }$ the limiting value of G(τ) for τ →∞. ω_0_ is the beam waist and describes the extent of the effective detection volume in the focal plane where the intensity has dropped to e^−2^.

In particular, an initial removal of the first 180,225 lines of each scan reduces the influence of photo bleaching. The analysis is further corrected for bleaching via a moving average. A high pass filter reduces slow fluctuations by dividing the total time into smaller pieces of about 36:25 s.

The autocorrelation curves are normalized to their G(0). In cases where no correlation is extracted, the curves are normalized to their maximum value (i.e., highest peak).

## Figures and Tables

**Figure 1 ijms-23-10405-f001:**
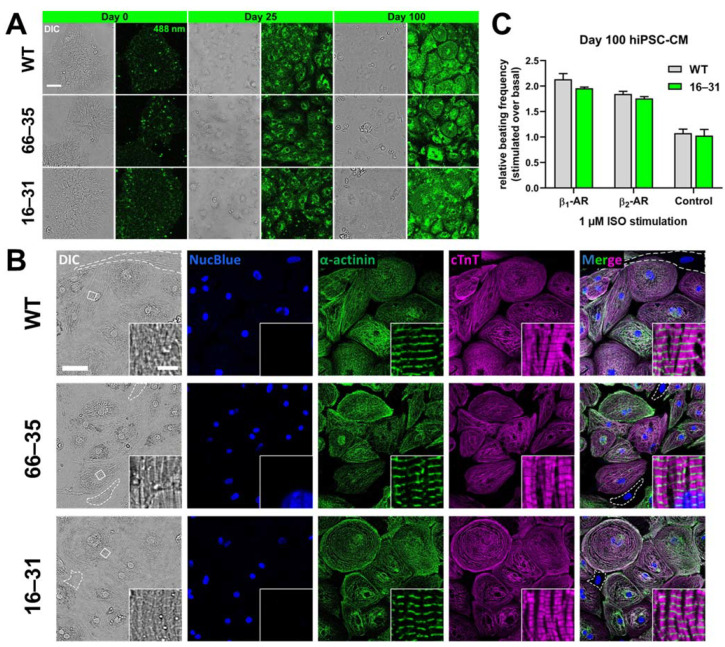
Structural and functional assessment of hiPSC-CMs. (**A**) Comparison of undifferentiated (day 0) and differentiated (day 25 and 100) WT hiPSC-CM and CRISPRed *HA-mEGFP-ADRB2* clones 66–35 and 16–31. Fluorescence images represent autofluorescence in the green spectral range at 488 nm. Shown are representative images of 2 replicates with imaging of ≥4 points-of-view. Contrast settings are identical. Scale bar is 50 µm. (**B**) Immunofluorescence staining of day 100 WT hiPSC-CM and clones 66–35 and 16–31 for typical cardiomyocyte markers α-actinin and cardiac troponin T (cTnT). Nuclei were stained with NucBlue. Insets show the magnifications of regions marked with white rectangles in DIC. Dashed lines in DIC and merge indicate cells lacking a cardiac phenotype, most likely fibroblasts. Representative images of 3 replicates with the imaging of ≥3 points-of-view are shown. Scale bar is 50 µm and in magnification 5 µm. (**C**) Functionality recording of WT hiPSC-CMs and clone 16–31 after 100 days of differentiation. Shown is the increase in beating frequency normalized to basal frequency upon stimulation with 1 µM isoproterenol (ISO) of β_1_-ARs and β_2_-ARs. The frequency remains unchanged upon stimulation after blocking the receptors with 10 µM propranolol (Control). Graph represents the mean ± SEM of ≥3 independent experiments.

**Figure 2 ijms-23-10405-f002:**
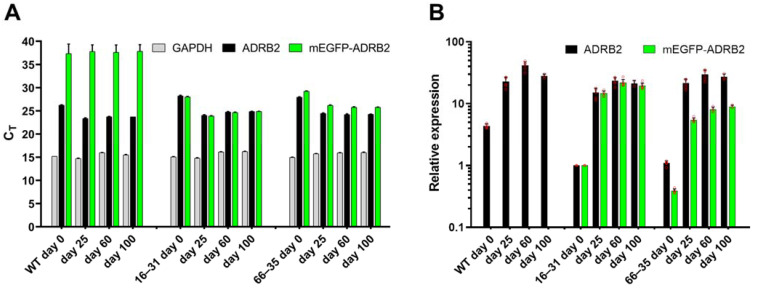
Indication of endogenous *ADRB2* and *mEGFP-ADRB2* mRNA expression during hiPSC-CM culture maturation. (**A**) Absolute cycle threshold (Ct) expression values for *GAPDH*, *ADRB2* and *mEGFP-ADRB2* are shown. (**B**) Relative expression of data shown in (**A**) normalized to *GAPDH* and calibrated to day 0 of clone 16–31. Data are mean ± SD of (3–6 replicates of) 2 independent experiments.

**Figure 3 ijms-23-10405-f003:**
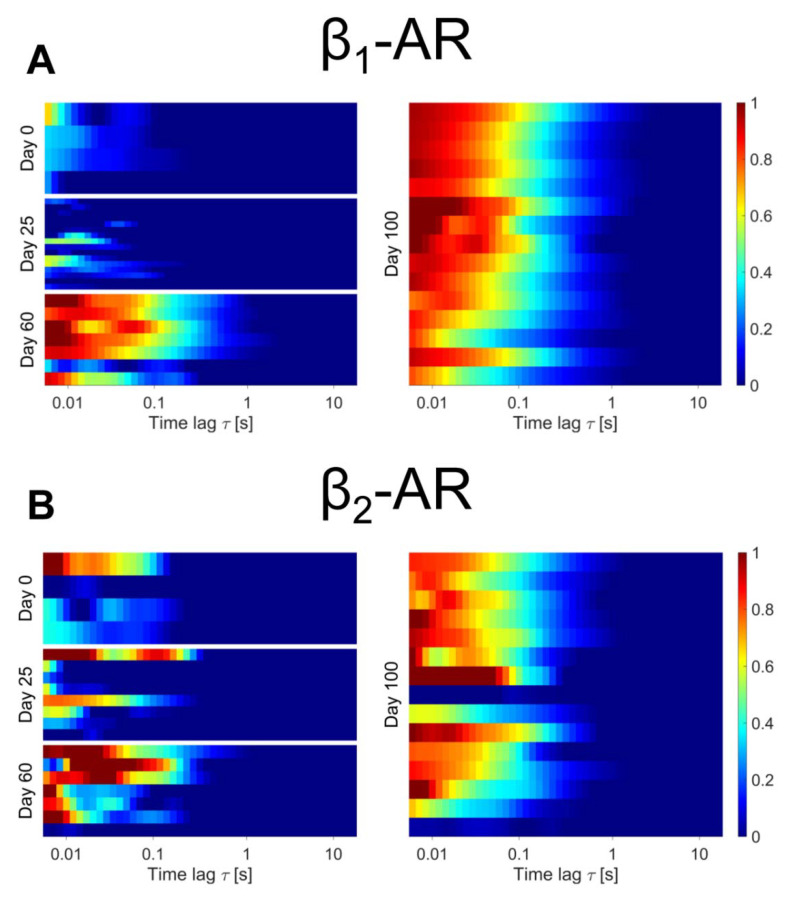
Linescan-FCS evaluation of β_1_-ARs (**A**) and β_2_-ARs (**B**) in WT hiPSC-CMs at day 0, 25, 60 and 100 of maturation. Colorscale indicates the amplitude of normalized autocorrelation (ACF) curves. Normalization was to each G(0) or highest peak of the curve (See Materials and Methods Section). Each bar represents the average correlation arising from the scan at the surface of one cell. For each day a minimum of at least four scans of different hiPSC-CMs was evaluated (n for β_1_-/β_2_-AR at day 0 = 4/4, day 25 = 16/8, day 60 = 7/7, day 100 = 15/15).

**Figure 4 ijms-23-10405-f004:**
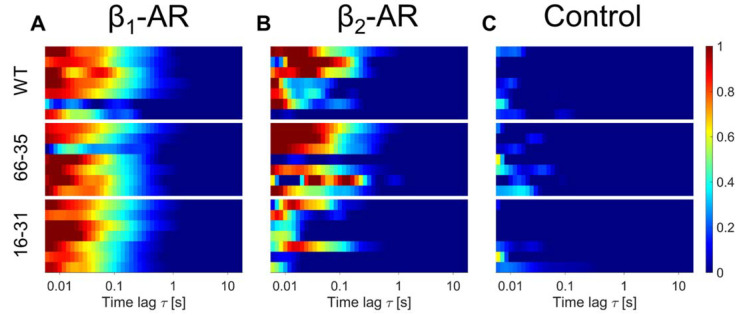
Comparison of β_1_-AR (**A**) and β_2_-AR (**B**) membrane expression in hiPSC-CMs at day 60 of maturation. (**C**) Shows the acquired linescan-FCS data of hiPSC-CMs at day 60 of maturation pre-treated with 100 µM propranolol (Control). Each subpanel represents wildtype (WT) hiPSC-CMs (**top**), clone 66–35 (**middle**) and clone 16–31 (**bottom**). Colorscale indicates the amplitude of normalized autocorrelation curves. Normalized was to each G(0) or highest peak of the curve (See Materials and Methods Section). Each bar represents the scan at the surface of one cell. For each cell line and condition scans of seven different hiPSC-CMs are evaluated. Both receptor subtypes show a diffusional fingerprint at the surface plasma membrane of the WT cells and clones at day 60. Control cells indicate no detected diffusing receptors (blue).

**Table 1 ijms-23-10405-t001:** List of crRNAs, HDR donors and primers used for genotyping and qRT-PCR.

Name	Sequence from 5′ to 3′	Description
crRNA- 103forw	CCTGCCAGACTGCGCGCCAT	crRNA used with generic tracrRNA to build the gRNA for Cas9 targeting to *ADRB2* locus. Binds over start codon for knock-in of mEGFP
Donor plasmid for knock-in of mEGFP	AAGCTTAACGGGCAGAACGCACTGCGA AGCGGCTTCTTCAGAGCACGGGCTGGAACTGGCAGGCA CCGCGAGCCCCTAGCACCCGACAAGCTGAGTGTGCAGGACGA GTCCCCACCACACCCACACCACAGCCGCTGAA TGAGGCTTCCAGGCGTCCGCTCGCGGCCCGCAGA GCCCCGCCGTGGGTCCGCCCGCTGAGGCGCCCCCAGCCAGTGCGCTCA CCTGCCAGACTGCGCGCCATGGTGAGCAAGGGCGAGGAGC TGTTCACCGGGGTGGTGCCCATCCTGGTCGAGCTGGACGGCGA CGTAAACGGCCACAAGTTCAGCGTGTCCGGCGAGGGCGAG GGCGATGCCACCTACGGCAAGCTGACCCTGAAGTTCATCTGCA CCACCGGCAAGCTGCCCGTGCCCTGGCCCACCCTCGTGAC CACCCTGACCTACGGCGTGCAGTGCTTCAGCCGCTACCCCGAC CACATGAAGCAGCACGACTTCTTCAAGTCCGCCATGCCCG AAGGCTACGTCCAGGAGCGCACCATCTTCTTCAAGGACGACGG CAACTACAAGACCCGCGCCGAGGTGAAGTTCGAGGGCGA CACCCTGGTGAACCGCATCGAGCTGAAGGGCATCGACTTCAAGG AGGACGGCAACATCCTGGGGCACAAGCTGGAGTACAACTA CAACAGCCACAACGTCTATATCATGGCCGACAAGCAGAAGAAC GGCATCAAGGTGAACTTCAAGATCCGCCACAACATCGAGG ACGGCAGCGTGCAGCTCGCCGACCACTACCAGCAGAACACCCC CATCGGCGACGGCCCCGTGCTGCTGCCCGACAACCACTAC CTGAGCACCCAGTCCAAGCTGAGCAAAGACCCCAACGAGAAGC GCGATCACATGGTCCTGCTGGAGTTCGTGACCGCCGCCGG GATCACTCTCGGCATGGACGAGCTGTACAAGtctagaGGGCAA CCCGGGAACGGCAGCGCCTTCTTGCTGGCACCCAATAGAA GCCATGCGCCGGACCACGACGTCACGCAGCAAAGGGACGAGGT GTGGGTGGTGGGCATGGGCATCGTCATGTCTCTCATCGTC CTGGCCATCGTGTTTGGCAATGTGCTGGTCATCACAGCCATTG CCAAGTTCGAGCGTCTGCAGACGGTCACCAACTACTTCAT CACTTCACTGGCCTGTGCTGAATTC	Donor sequence (1.2 kbp) encoding for mEGFP plus 230 bp homology arms flanked by HindIII (5′) and EcoRI (3′) restriction enzyme sites, which were used for cloning into a pcDNA3.1(+) vector backbone. Served as template to generate the partially single-stranded hybrid HDR donor cocktail for knock-in of mEGFP at the *ADRB2* locus using the following 4 primers: mEGFP_N-term_f mEGFP_C-term_linker_r b2AR-hum_120bp_5′arm_f1 b2AR-hum_120bp_3′arm_r1
crRNA- 69forw	CTGCGCGCCATGGTGAGCAA	crRNA used with generic tracrRNA to build the gRNA for Cas9 targeting to the *mEGFP-ADRB2* locus. Binds over start codon for knock-in of HA-SS
ssODN for knock-in of HA-SS	TGGGTCCGCCCGCTGAGGCGCCCCCA GCCAGTGCGCTCACCTGCCAGACTGCGCGCCATGA AGACGATCATCGCCCTGAGCTACATCTTCTGCCTGGTA TTCGCCGTGAGCAAGGGCGAGGAGCTGTTCACCGG GGTGGTGCCCATCCTGGTCGAGCTGGACGGC	ssODN consisting of 60 bp homology arms used as HDR donor template for knock-in of HA signal sequence at the N-terminus of *mEGFP-ADRB2* locus.
mEGFP_ N-term_f	ATGGTGAGCAAGGGCGAGGA	Forward primer for generation of partially single-stranded hybrid HDR donor, binds N-terminus of *mEGFP*
mEGFP_ C-term_ linker_r	tctagaCTTGTACAGCTCGTCC	Reverse primer for generation of partially single-stranded hybrid HDR donor, binds C-terminus of *mEGFP* including a linker sequence
b2AR-hum_120bp_ 5′arm_f1	CCCACACCACAGCCGCTGAA	Forward primer for generation of partially single-stranded hybrid HDR donor, binds to 5′UTR of *ADRB2*
b2AR-hum_120bp_ 3′arm_r1	AGACATGACGATGCCCATGC	Reverse primer for generation of partially single-stranded hybrid HDR donor, binds to N-terminus of *ADRB2*
b2AR-hum_ mEGFP- ADRB2_f1	ATTGGCCGAAAGTTCCCGTA	Forward sequencing primer: verification of the endogenous insertion of HDR donor template at the *ADRB2* locus, binds to 5′UTR of *ADRB2* upstream of 5′-homology arm of HDR donor
b2AR-hum_ mEGFP- ADRB2_r1	GTCCAAAACTCGCACCAGAA	Reverse sequencing primer: verification of the endogenous insertion of HDR donor template at the *ADRB2* locus, binds to N-terminus of *ADRB2* downstream of the 3′homology arm of HDR donor
2nd qPCR_ GFP_Fwd	TGAGCAAAGACCCCAACGAG	Forward qPCR primer for amplification of *mEGFP-ADRB2*. Binds to C-terminus of *mEGFP*
2nd qPCR_ ADRB2_Rev	CGCATGGCTTCTATTGGGTG	Reverse qPCR primer for amplification of *mEGFP-ADRB2*. Binds to N-terminus of *ADRB2*

## Data Availability

The data presented in this study can be made available upon request.

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
