# Peer review of "Fluorescence Spectroscopy of Low-Level Endogenous β-Adrenergic Receptor Expression at the Plasma Membrane of Differentiating Human iPSC-Derived Cardiomyocytes"

_ijms, 2022, doi:10.3390/ijms231810405_

Round 1
Reviewer 1 Report
Gmach et.al. have looked at the expression and localization of β1- and β2-adrenergic receptors (β1/2-ARs), during hiPSC- cardiomyocyte differentiation at various time points. the authors have employed linescan Fluorescence Correlation Spectroscopy (FCS) a technique that has previously been used by other researchers as well ( https://doi.org/10.1016/j.tips.2007.09.008, https://doi.org/10.1096/fj.202002268R). The authors show that during the various stages of hiPSC differentiation into cardiomyocytes there is a spatio-temporal shift in the expression of β1/2-ARs. Although the concept of this study is indeed interesting there are several concerns.
Major comments:
Fig 1- why did the authors include 66-35 heterozygous clone in this analysis. what was the rationale behind it? there seems to be no added information arising from the use of this clone in the manuscript.
Fig 1- the authors claim that there is a strong increase in autofluorescence between pluripotent state to cardiomyocyte (at day 100 of differentiation). However the culture conditions are completely different and the changes in autofluorescence might be caused due to the technical differences in medium, coating, pH etc. Therefore fig 1A seems irrelevant.
The authors discuss about the lack of a complete T-tubular network in hiPSC-CMs at day100. Since this study was aimed at identifying the localization of the receptors did they consider maturing the hiPSC-CMs using other media compositions (10.1161/CIRCRESAHA.117.311920, https://doi.org/10.1038/s41467-021-23816-3 ) ? Did they check how mature the T-tubule network is in their day 100 hiPSC-CM using electron microscopy? This is crucial since the FCS data shown is comprised of only 4 cells and the high degree of heterogeneity in hiPSC-CMs is also well documented even at later timepoints of differentiation.
The shift in ct values for eGFP are quite significant in the knock in cell lines. It is unclear why the authors cannot detect GFP levels in western blot? Have the authors considered a mass spec analysis to identify the abundance of GFP in their knock-in cell lines to prove that the observed effects are indeed due to the final protein being made in the cell?
The pluripotency characterization of the clones 16-31 and 66-35, created using CRISPR/Cas9 editing in this study are missing.
Fig 1C- The spontaneous beating frequency measurements are not reliable. The videos clearly show that the cells are not cultured in a single monolayer. HiPSC-CMs can have different beating rates depending on their seeding densities. Kindly consider a reliable MEA analysis to make claims regarding beating frequencies (10.1111/j.1476-5381.2011.01623.x )
Minor comments:
In several parts of the text 'CM-differentiated hiPSC' has been used which is misleading. kindly change it to 'CM-differentiated from hiPSC' of 'hiPSC-differentiated CMs'
please improve the methods section. for eg- it has not been described what the authors mean by CDM3 ?
Reviewer 2 Report
1. The authors cultured hiPSC cultured hiPSCs for up to 100 days. To my knowledge, hiPSCs in higher passage numbers are prone to have chromosome abnormalities. Have the authors checked this issue? If so, it is good to include this information, at least in the supplementary files.
2. In Figure 1B, the authors showed the sarcomeric structures ~100-day-old hiPSC-CMs. It is nice to see the authors used long-term cultured hiPSC-CMs rather than 30-day-old cells that are very immature. Can the authors also check the metabolism in these older cells, maybe a seahorse assay and/or mitochondrial staining? Because the metabolic switch from glucose to fatty acid is a key hallmark for the maturation of cardiomyocytes.
3. GAPDH was used as a housekeeping gene to normalize the expression of ADRB2 in WT hiPSC and the other two clones. Merging studies have pointed out the tissue- and cell-specific housekeep genes and GAPDH mRNA levels might not be the best option for cardiomyocytes/hearts. Did the authors consider other housekeeping genes or include several ones in this case?
4. In Figure 2, it would be nice to show the bar graph with overlapping dots representing each individual value per group.
5. In Figure 3, the authors showed higher cell-to-cell heterogeneity in WT hiPSC-CMs on day 0 and day 25 when compared to cells on day 60. Did the authors also observe this variation morphologically and can the authors elucidate more into this finding?
6. Besides plasma membrane, ARRB1 and ARRB2 are also highly expressed in the nucleus, lysosome, etc. How did the authors exclude signals coming from other cellular organelles when detecting and quantifying β2-AR expression levels at the plasma membrane?
Round 2
Reviewer 1 Report
Firstly, the comment regarding autofluorescence was answered by the authors but unfortunately in a completely different aspect. The comment was made specifically regarding why they are showing/comparing Day 0 (undifferentiated-pluripotent) vs Day 100 (differentiated –cardiomyocyte state). If the aim is to indicate maturity then it is better to show mitotracker staining /autofluorescence of cardiomyocytes from day 25 vs day 100 instead of comparing hiPSCs to hiPSC-CMs. Comparing hiPSC to hiPSC-CMs indicates successful differentiation but definitely does not indicate how mature the cardiomyocyte became over the long-term culture conditions.
Secondly, this is great to know that the the hiPSC-CMs produced by the authors have ~2 μm sarcomere length without any specific treatment. Perhaps this can be added as additional data for characterization of hiPSC-CMs used in this study along with mitotracker.
Thirdly, the figure legends 3 and 4 only mention about “minimum of at least 4 scans” or “evaluates 4 cells”. If the authors have indeed analyzed 4 cells for day0 hiPSC, but at least 8 cells for the later time points, and 15 cells for day 90, they could have simply taken the effort to specify it properly in the manuscript. I would highly suggest the authors to mention correct ‘n’ numbers in figure legends and/or provide specific details in the methods section.
Fourthly, the idea (as described in the abstract) was to compare the localization of the β1- and β2-ARs based on their previous findings from adult CMs where the β2-ARs specifically segregated to T-tubules. The authors in this work, clearly discuss and show (rebuttal Fig 2E right) the absence of T-tubules in the hiPSC-CMs used in this study. Therefore it is better to re-phrase the abstract section to suit better to the data shown later on.
